# Entropy-based dynamic ensemble classication algorithm for imbalanced data stream with concept drift

**JiaMing Gong** [1,3]*, **MingGang Dong**[2]

**1** College of Data Science, Guangzhou Huashang College, Guangzhou, Guangdong, China, **2** College of Information Science and Engineering, Guilin University of Technology, Guilin, Guangxi, China, **3** St. Paul University Philippines, Province of Cagayan, Tuguegarao City, Philippines

* hsjm9616@qq.com

**Data Availability Statement:** All relevant data are within the manuscript and its Supporting information files.

**Funding:** This work was supported by the Research on Dynamic Weighted Ensemble

## Abstract

Online imbalanced learning is an emerging topic that combines the challenges of class imbalance and concept drift. However, current works account for issues of class imbalance and concept drift. And only few works have considered these issues simultaneously. To this end, this paper proposes an entropy-based dynamic ensemble classification algorithm (EDAC) to consider data streams with class imbalance and concept drift simultaneously. First, to address the problem of imbalanced learning in training data chunks arriving at different times, EDAC adopts an entropy-based balanced strategy. It divides the data chunks into multiple balanced sample pairs based on the differences in the information entropy between classes in the sample data chunk. Additionally, we propose a density-based sampling method to improve the accuracy of classifying minority class samples into high quality samples and common samples via the density of similar samples. In this manner high quality and common samples are randomly selected for training the classifier. Finally, to solve the issue of concept drift, EDAC designs and implements an ensemble classifier that uses a self-feedback strategy to determine the initial weight of the classifier by adjusting the weight of the sub-classifier according to the performance on the arrived data chunks. The experimental results demonstrate that EDAC outperforms five state-of-the-art algorithms considering four synthetic and one real-world data streams.

## Introduction

Online imbalanced learning has drawn considerable interests from both academia and industry. It aims to tackle data streams with skewed class distributions and concept drift. These problems commonly exist in real-world applications, such as sewage disposal, traffic control, spam filters, and electronic commerce [1]. Since such data streams are massive, dynamic, real-time, and high speed [2], they can be read-only once by an application [3]. This introduces the issue of classifying imbalanced data with concept drift [4] to online learning [5]. However, most standard algorithms consider the problem of balanced class distributions. Therefore, when the data stream becomes complex and imbalanced classification becomes inaccurate as majority classes are favored by classifiers [6].

Classification Algorithm for Imbalanced Data Streams Award Number:2023HSDS16. The funders played a significant role in our study, specifically in the preparation of the manuscript and the decision to publish.

**Competing interests:** The authors have declared that no competing interests exist.

To improve the accuracy of classifieries for imbalanced data, the sampling technique is one of the most effective approaches. Generally, the sampling technique can be divided into oversampling and undersampling [7, 8]. In oversampling, SMOTE is the representative approach for imbalanced data, first proposed in Ref. [9]. SMOTE leverages minority attributes to generate new samples, thereby balancing the amount of majority and minority attributes in samples. Several approaches based on SMOTE, have been proposed, such as borderline-SMOTE [10], ADASYN [11], REA [12], MWMOTE [13] and AMDO [14]. Additionally, Li et al., have proposed an entropy differences based oversampling approach for imbalanced learning [15]. The work in [16] presents a dynamic feature group method for imbalanced data distributions. It extracts group characteristics of samples and re-balanced distributions via the importance sampling technique. The authors of [17] employed a selectively recursive approach algorithm. By selectively placing the previously received positive samples into the current data chunk, the algorithm can significantly improve the accuracy at the expense of sample space. However, considering real-time and high speed data streams, these SMOTE-based techniques are not suitable in that they are time-consuming as they increase the amount of minority samples. Recognizing the critical demand for real-time analysis of data chunks in dynamic streaming environments, Monika et al. introduced a novel hybrid algorithm [18] that fuses the strengths of FFSMOTE and the bee algorithm. This eliminates the need for periodic analyses and model rebuilds, allowing for instantaneous updates as new data arrives.

In the undersampling technique, the same amount of minority and majority samples are selected. The advantage of undersampling is that it significantly decreases the computational time by removing the majority samples. However, using undersampling technique alone substantially deteriorates in the accuracy due to the low information content. To address these issues and enhance the performance, undersampling is generally combined with ensemble classifiers that leverage multiple sub-classifiers to make decisions. Ditzler et al. have proposed two ensemble approaches, Learn++.CDS and Learn++.NIE [19], which share similar frameworks. Initially, these models introduce a certain amount of data into a data chunk from the data stream. Then, the sub-classifier can be created for each arrived data chunk, and the arrival time is recorded as the creation time. In this manner, a new data chunk is generated from the previous by the sub-classifiers. Each sub-classifier can be assigned a weight that linearly increases with the with creation time for each data chunk. Due to the use of sophisticated multiple sub-classifiers and weights, this framework is capable of realizing high accuracy and efficiency, and is applied widely.

RUEO [20] utilizes an ensemble learning strategy, which combines multiple base classifiers into a single, more robust classifier. This approach harnesses the strengths of individual classifiers while mitigating their weaknesses, particularly in the face of complex and dynamic data streams.

Based on a similar framework, Wang et al., employed a time decay function to capture the imbalance rate dynamically and proposed two learning algorithms, UOB and OOB [21], with the aim of adjusting the sample distribution to guarantee sample balance. UOB outperforms benchmarks in terms of the recognition accuracy for minority samples, and OOB is robust against dynamic changes in the class imbalance. Subsequently, these were improved further to develop and implement WEOB1 and WEOB2 [22]. In [23], the integrated learning framework has been leveraged to address the problem of imbalanced data streams. It was demonstrated that ensemble learning can solve the problem of cost-sensitive learning and online learning simultaneously. The above works have combined sampling techniques and ensemble learning to solve the problem of imbalanced data stream. However, these approaches have limited performance on imbalanced data streams with issue of concept drift [24].

MicFoal [25] efficiently classifies multiclass imbalanced network traffic with concept drift by selectively requesting labels for uncertain samples based on a variable threshold and adaptively adjusting the label budget. It addresses the imbalance and drift issues through a sample weighting formula and modular design for flexibility.

To solve the concept drift problem, Du H et al., have designed and implemented a DWM algorithm [26], which is a general framework based on the weighted summation algorithm of the base classifier. In DWM, the rule for classification on the base classifier is maintained in an unchanged state. Moreover, the weight of the base classifier is dynamically trained and modified on each arrival data accordingly. However, this method causes classifier redundancy when modifying the classifier weights alone. To avoid increasing the classifier redundancy, the authors of [27] applied a dynamic weighted majority for incremental learning algorithm (DWMIL). This method divides data chunks into multiple sub-sample sets by undersampling. Then, it creates a classifier for each data chunk and continuously adjusts its weight according to the performance of the classifier in the subsequent data chunk while eliminating weights below a certain value. Thereby DWMIL yields better performance in improving the accuracy of the classifier, as well as increasing the execution efficiency of the algorithm. Using mathematical methods, Ghazikhani et al. established an online perception model [28] to resolve the problem of imbalanced data stream and concept drift. In their work, the class imbalance problem is solved by adjusting the weighted error of the base classifier, and the issue of concept drift in data streams is processed by a recursive least squares error model, respectively. Furthermore, considering concept drift, under the use of parallel ELM-based ensemble classifier in [29], UELM-MapReduce has been proposed to dynamically adjust the weight of each base classifier by removing the classifiers with low accuracy. Similarly, Learn++.NSE [30] trains a new classifier for each received data chunk and combines these classifiers with a dynamic weighted classifier. This allows identifying changes in the distribution of basic data and implementing actions to adapt to concept drift [31].

LEPID [32] adopts a selective and prioritized approach in identifying and emphasizing the significance of concept drift prototypes and minority class prototypes by assigning them augmented weights. This strategy directs attention towards regions that most accurately encapsulate the current conceptual landscape, enhancing the algorithm's responsiveness to changes in the data stream.

In summary, considering concept drift in imbalanced data streams, these algorithms have focused significantly on the amount of samples by neglecting the importance of the information and quality of the samples. This can lead to low recognition accuracy for minority samples. The classifier weights are initialized to fixed values, which can-not enhance the classifier performance to cope with the issue of concept drift [33].

To address these limitations, we propose an entropy-based dynamic ensemble classification EDAC algorithm for imbalanced data streams with concept drift. Compared to previous works, the main difference of EDAC is that it takes into account the entropy of the samples, which indicates the information carried by the samples and the quality of the samples. The general process of EDAC is as follows. Initially, according to the amount of entropy carried by the entire data chunk, EDAC divides the data chunk into multi-pairs of balanced samples. Next, to enhance the accuracy of the classifier, EDAC selects high quality samples to participate in the training of the classifier. Additionally, the weight on the classifier is initialized based on the performance on each classifier. And the weight is dynamically updated based on the performance on the new data chunk. We conduct simulations to verify the performance of EDAC on four synthetic and one real-world data stream [34]. The experimental results show that EDAC outperforms the-state-of-the-art algorithms [35].

In summary, the contributions of the proposed EDAC can be highlighted as follows:

- To balance the imbalanced data stream, EDAC presents an entropy-based balanced strategy, because the entropy can reflect the critical information of samples. The entropy is calculated based on probability theory. EDAC divides imbalanced data chunks into multiple balanced sample pairs. Each sample pair is trained with one classification rule, and multiple classification rules form the classifier of the current data chunk. In this manner EDAC yields better performance in improving the accuracy of classification.

- To enhance the classification of minority samples, EDAC introduces a density-based sampling method. Specifically, the strategy categorizes minority instances into high quality and common instances in terms of the density of similar samples. During this process, the strategy selects the amount of minority instances with the highest quality instances. Additionally, the common samples are randomly selected within a slight amount. In this manner, the amount of majority and minority instances can balanced while using the technique of undersampling.

- To address the issue of concept drift, EDAC also adopts the idea of ensemble classification to develop an approach for dynamically setting the weight on each classifier. Initially, the base classifier is created based on current data chunk, so the initial weight of the base classifier can be calculated through the current data chunk. Then, the initial weight and base classifier can be calculated an updated by using the subsequent data chunks. When the weight is less than the threshold, it indicates that the concept on that classifier is not suitable for the new data chunk, and thus, the associated classifier must be removed. Each data chunk has a base classifier with a weight that fits into the concept of the forthcoming data chunk.

- To verify the effectiveness and performance of EDAC, a simulation was conducted, and the performance was compared with those of five state-of-the-art algorithms, REA, ADAB, MWMOTEB, DWMIL, and DWM. IN the simulation, four synthetic data streams: SEA, Moving Gaussian, Hyper Plane, Checkerboard, and one real-world data stream, Electricity, were used The experimental results demonstrate the proposed algorithm outperforms the other algorithms in terms of the AUC, G-mean, and recall.

## The proposed algorithm: EDAC

### The overview of the EDAC algorithm

In this subsection, we provide the overview of EDAC. The framework of the proposed EDAC algorithm is shown in Fig 1. We assume that the data stream arrives in a chunk by chunk manner, and data chunk 1 is the first to arrive. We calculate the sample entropy value of each arriving data chunk to get the information content $Ei$ of the sample population. The data chunk is divided into $T$ equal subsets, where the sample information content of the positive samples in the subset is equal to that of the negative samples. Then the quality of each minority class instances of the data chunk is calculated. Next, minority samples are divided into high quality samples and common samples based on the the quality of samples. When selecting minority class instances, EDAC selects all high quality instances, and the common instances are randomly selected to make the total number of selected instances equal to the number of minority class instances in the overall data chunk. In the Fig 1, the red line {$Line_1$, $Line_2$, ..., $Line_i$} represents the threshold $\theta$. The instances below the threshold are the sample with low quality, and the instances above the threshold are the sample with high quality. This minority class sample above the threshold carries more information and can better represent the minority sample groups in the data block; therefore, we try to select the minority class sample points above the

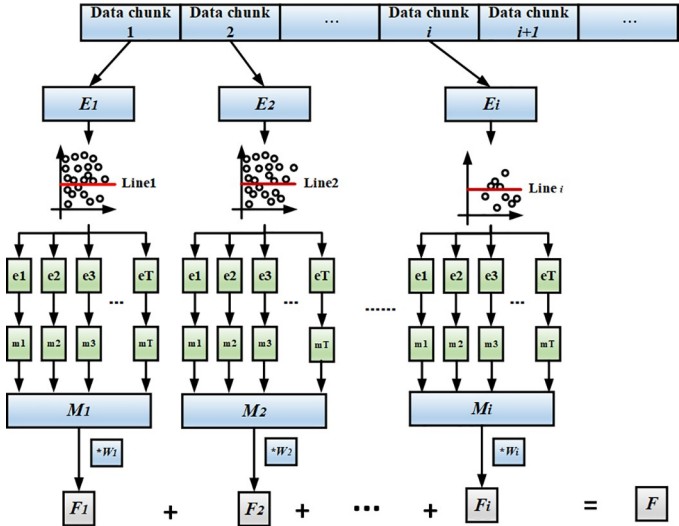

**Fig 1. Framework of EDAC.**

threshold. We not only select these sample points with high information content, which will cause overgeneralization; but also randomly select the minority class sample points below the horizontal line to make the minority class sample more reasonable.

EDAC selects the same number of majority samples randomly as the number of minority samples to make the subset in equivalent to the original set. Then, according to the obtained equilibrium samples for {$E1$, $E2$, $E3$, . . ., $Ei$}, training classification rules {$M1$, $M2$, $M3$, . . ., $Mi$} are respectively formulated. The classifier Mi of the current data chunk comprises of {$m1$, $m2$, $m3$, . . ., $mT$}. The initial weight $Wi$ of each classifier is determined by the performance of the classifier in the data chunk that arrives later. Finally, the prediction result $F_{(i-1)}$ is weighted by $W_{(i-1)}$ to obtain the final prediction result $Fi$ on the current testing data chunk $i$.

## Entropy-based balanced strategy

Normally, the imbalance ratio of a data chunk is calculated by dividing the large samples by minority samples. However, it can not describe the information content of different samples. EID [36] measures the information content of different samples and calculates the imbalance rate in terms of the difference between them. This manner of introducing the effective information content when calculating an imbalanced ratio can significantly improve the robustness of classifiers in imbalanced problems. Therefore, we propose entropy-based balanced strategy, that introduces the effective entropy of the samples to balance the data chunk.

For each data chunk, we calculate the entropy of each class. Entropy represents the expected amount of information for the corresponding category. Then, we calculate the entropy of the current data chunk and divide the current data chunk into T subsets by the entropy of minority class. The problem of undersampling, which removes examples from the majority class, could cause missing concepts pertaining to the majority class of the classifier. Therefore, for each balanced sample pair divided by the entropy, we first fixed high quality minority class samples and select random common minority class samples, based on the number of minority class samples to improve the accuracy. To further strengthen our approach, we implement an undersampling technique to randomly select an equal number of majority class samples for each subset. These subsets are then used to train individual sub-classifiers, which collectively

form the ensemble classifier for the given data chunk. By aggregating multiple classification rules, we mitigate the potential loss of majority class concepts, thereby enhancing the overall performance and robustness of our classification model. The specific process is as follows:

$$\rho(x_i) = \frac{1}{q} \cdot \sum_{c=1}^{q} dist(x_i, s(x_i)_c), 0 < q \le k, \tag{1}$$

$\rho(x_i)$ represents the average distance between $x_i$ and its intra-class $q$ neighbors.

$$\omega(x_i) = \frac{\rho(x_i)}{\sum\limits_{c=1}^{N} \rho(x_i)}, i \in 1, 2, \ldots, N_n, \tag{2}$$

$\omega(x_i)$ represents the percentage of $x_i$ in the current data chunk instances density, it can be regarded as the probability for $x_i$ in the overall instance.

$$S_{maj} = -\sum_{i=1}^{N_{maj}} \omega(x_i) \cdot \log_2 \omega(x_i), i \in 1, 2, \ldots, N_n, \tag{3}$$

$$S_{min} = -\sum_{j=1}^{N_{min}} \omega(x_j) \cdot \log_2 \omega(x_j), j \in 1, 2, \ldots, N_p, \tag{4}$$

$$T = \frac{S_{min} + S_{maj}}{2 \cdot S_{min}}, \tag{5}$$

$S_{maj}$, $S_{min}$ are the entropy values of the majority and minority samples, respectively. We adopt the strategy of undersampling based on the information of minority class samples and divide the overall entropy value of samples by the entropy value of minority class samples as the basis of divided data chunks. It can be known that the information content of the entire data block is divided into $T$ subsets, and each subset represents the amount of partial information contained in the current data chunk. For each subset, we select the same number of majority class instances as the number of minority class instances to make the subset reach equilibrium. The pseudo-code of dividing data chunk is given in Algorithm 1:

**Algorithm 1** Entropy-based balanced strategy

```
Input: Data chunk at timestamp t: D^(t) = {x_i ∈ X, y_i ∈ Y}, i = 1, 2, ...,
       N; The positive instances of D^(t) : Dp_c^t, c = 1, 2, ..., N_p; The negative
       instances of D_c^(t) : Dn^(t), c = 1, 2, ..., N_n;
Output: the number of dividing data chunk: T
1: Calculate the average distance ρ_(x_i) between x_i and its intra-class q
   neighbors using Eq (1)
2: Calculate the percentage ω_(x_i) for x_i in the current data chunk using
   Eq (2)
3: for j ← 1 to N_p do
4:    Smin ← sum(ω_xj · log_2 ω_xj)
5: end for
6: for i ← 1 to N_n do
7:    Smaj ← sum(ω_xi · log_2 ω_xi)
8: end for
9: T = (Smin+Smaj)/(2*Smin)
```

## Density-based sampling method

Although our selection strategy can efficiently avoid over-fitting and over-generation via random and disordered selection of samples, this type of selection without a strategy may reduce the precision. The classification effect of imbalanced data has low performance, because the recognition accuracy of minority class samples is not high. To improve the recognition accuracy of minority class samples, Gao proposed to collect all previous minority class samples to amplify the current training data chunk, and adopted undersampling to select samples [37]. However, this approach can expand the sample space but may not guarantee the selection of informative samples essential for classifier training. In contrast, methods like SERA and REA selectively incorporate minority class samples that exhibit correlations with the current minority samples, leading to superior prediction performance. Inspired by this, we introduce a density-based sampling method that assesses the quality of minority class instances based on the density of similar samples within each data chunk. By distinguishing between high-quality and common instances, we prioritize the selection of high-quality samples, which are likely to be core instances representative of the minority class. The specific process involves calculating the information content of minority samples by assessing their density-based quality. This approach ensures that the selected samples contribute significantly to improving the recognition rate of minority class instances, thereby enhancing the overall performance of the classifier. The specific process of calculating the information content of minority samples is as follows. For each sample $x_i \in \{c_{maj}, c_{min}\}$, $i = 1, 2, \ldots, n$, to obtain the density of $x_i$, divide the number of instances and intra-class neighbors by its average distance.

$$density(x_i) = q \cdot \sum_{c=1}^{q} \frac{1}{dist(x_i, s(x_i)_c)}, \tag{6}$$

Here, $q$ is the number of nearest intra-class neighbors in $k$ nearest neighbors of $x_i$, $s(x_i)$ are nearest intra-class neighbors of $x_i$, $dist(x_i, s(x_i)_c)$ measures the distance between $x_i$, and $s(t)$, $\{t = 1, 2, \ldots, p\}$ are nearest intra-class neighbors. We take the ratio of the number of instances which is the same type in the $k$ nearest neighbors of sample $x_i$ to the distance from $x_i$ to $s(x_i)$ without other class instances to describe the density of $x_i$. We only consider the distribution relationship between the same class instances over the whole data set. The closer the same class instances around $x_i$, the more the samples of the same class instances around $x_i$ greater the density of the sample $x_i$, the more it can reflect that sample $x_i$ is a core point, carrying more important information contents. If $p = 0$, there are no same class instances around $x_i$, which may affect the classification of the classifier as a noise sample.

Then, we calculate the intra-class density based on the density of each minority sample point to express the amount of information carried.

$$m(x_i) = \frac{density(x_i)}{\sum_{i=1}^{n} density(x_i)}, i \in 1, 2, \ldots N_{min}, \tag{7}$$

Here, $m(x_i)$ represents the percentage of $x_i$ in the overall minority class instances density, which estimates the quality of $x_i$. $N_{min}$ is the number of minority instances on the current data chunk. It is known that the higher the sample density for $x_i$, the more density-based information content it carries. We assume that the data stream arrives in chunk mode; the current entire data chunk is regarded as a training set, and the next data chunk to arrive as a test set. There are only two types of identifications in the data stream: positive type and negative type. The proportion of positive type samples in each data chunk is less than that of negative type

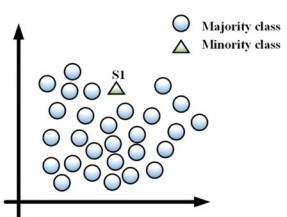 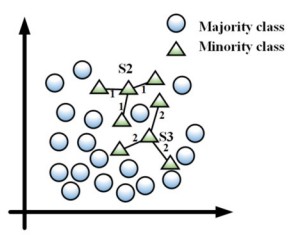 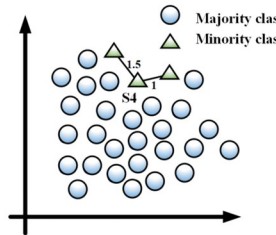

Minority samples division(a)    Minority samples division(b)    Minority samples division(c)

**Fig 2. Minority samples division.**

samples; Let $t$ denote a timestamp, $D^{(t-1)}$ denote the data chunk arrived at the time stamp $(t-1)$, and the data chunk arrived at the next time is $D^{(t)}$ Then, the data stream is $\{\ldots, D^{(t-1)}, D^{(t)}, D^{(t+1)}, \ldots\}$. A pseudo-code of the sampling method is thus formulated as follows:

**Algorithm 2** Density-based sampling method

```
Input: Data chunk at timestamp t: D⁽ᵗ⁾ = {xᵢ ∈ X, yᵢ ∈ Y}, i = 1, 2, ...,
       N; The positive instances of D⁽ᵗ⁾: Dpᶜᵗ, c = 1, 2, ..., Nₚ; The negative
       instances of Dᶜ⁽ᵗ⁾: Dn⁽ᵗ⁾, c = 1, 2, ..., Nₙ; The threshold to determine
       sample quality is θ;
Output: classification model hᵢ
1: Calculate the density den(c) of each minority sample point using
   Eq (6)
2: for i ← 1 to Nₚ do
3:     m(i) = den(i) / Σᴺᵖ₁=ᶜ den(c)
4:     if m(i) > θ then
5:        HP ← m(i)
6:     else
7:        CP ← m(i)
8:     end if
9: end for
10: Dₙ ← bootstrap Nₚ negative instances from Dnᶜ⁽ᵗ⁾
11: ins ← bootstrap Nᶜp instances from CP
12: Dₚ ← HP + ins
13: hₜ = Base_Learner(Dₙ, Dₚ)
```

As shown in Fig 2, the circles denote the current data chunk minority class samples, and the triangles represent current data chunk majority class samples. We assume that the $K$ value of the nearest neighbor is 3, and the threshold to determine sample quality $\theta$ is 0.6. S1 is an outlier, where its three nearest neighbors are another class and $m(S1) = 0$. In Fig 2(a), $m(S2) = \frac{3}{1+1+1} = 1 > \theta$ Fig 2(b), $m(S3) = \frac{3}{2+2+2} = 0.5 < \theta$ Fig 2(b), $m(S4) = \frac{2}{1.5+1} = 0.8 > \theta$ Fig 2(c), where $S2$, $S4$ are divided into high quality samples, and $S3$ is divided into common samples.

**Algorithm 3** EDAC

```
Input: Data chunk at timestamp t:D⁽ᵗ⁾ = {xᵢ ∈ X, yᵢ ∈ Y}, i = 1, 2, ...,
       N; The positive instances of D⁽ᵗ⁾: Dpᶜᵗ, c = 1, 2, ..., Nₚ; The negative
       instances of D⁽ᵗ⁾: Dnᶜ⁽ᵗ⁾, c = 1, 2, ..., Nₙ; The threshold of classificaer
       weight δ; the number of dividing data chunk: T
Output: prediction y′
1: When the first data chunk arrives, where t = 1, each instance y′ in
   the data chunk D⁽¹⁾ is tested directly.
```

```
 2: for t ← 2 to N do
 3:     y' = sign(∑_{l=1}^{m} w_j^{(t-1)} · H_j^{(t-1)}(x_t))
 4: end for
 5: for j ← 1 to m do
 6:    the error α_j^{(t)} was selected by using 1 – Gmean; update weight of
       base classifiers:
 7:     w_j^{(t)} = (1 - α_j^{(t)}) · w_j^{(t-1)}
 8: end for
 9: Create a new base classifier
10: m ← m + 1
11: for i ← 1 to T do
12:    H ← {h1, h2, ... hT}, where {h1, h2, ... hT} created by Algorithm 2
13: end for
14: The initialize weight of H is obtained by calculating its error
    rate ε_1 in the samples of current data chunk: w_m^{(t)} ← 1 - ε_1
15: Persist the classifier which the weight higher than δ:
16: H^{(t)} ← {H^{(t)}|w_j^{(t)} > δ}
17: m ← |H^{(t)}|
```

## Dynamic weight adjustment strategy for the ensemble classifier

Combining Algorithms 1 and 2, this subsection elaborates the EDAC and its integration with the with dynamic weight adjustment strategy. For each arriving data chunk $D_i$, we construct a new classifier $H_i$, where T denotes the number of dividing data chunk obtained by Algorithm 1. The subclassification rules $h_i$ are obtained by Algorithm 2. The initial weight $w_1$ is obtained by calculating its error rate $\epsilon 1$ in the samples of the current data chunk: $w1 = 1 - \epsilon 1$. We conduct a simple test of the newly created classifier to evaluate its classification performance. This self-feedback strategy can better reflect the situation when the classifier is generated. To reduce the influence of previous classifiers on new data stream samples and adapt to the newly arrived samples, the weight $w_1$ of the generated classifier is gradually reduced with the arrival of the new data chunk $D_{i+1}$. The error $\alpha_i$ can be calculated by evaluation functions such as G-mean, F-measure, precision, or recall. The formula for weight adjustment is:

$$w_i = (1 - \alpha_i) \cdot w_{i-1}, i \in 2, 3, \dots, n, \tag{8}$$

## Complexity analysis of EDAC

Through Algorithm 1, we can see the main complexity is the loop for calculating the entropy of samples. So, the complexity of Algorithm 1 can be bounded within $O(k \cdot n(n + 1))$. Algorithm 2 mainly to calculate the density of each minority sample. The complexity for Algorithm 2 is $O(k \cdot n(n - 1)) + O(k \cdot n)$; As we have analyzed above, we can find the complexity of EDAC largely depends on the step of Algorithm 1 $O(n(n + 1))$. Therefore, the complexity of algorithm 3 is $O(n(n + 1))$.

## Evaluation

This section is to give the setting of the algorithm in detail and describes the imbalanced data stream with concept drift used in the experiment, including four artificial data streams, one real-world data stream. Furthermore, we compare our proposed approach with five other methods to demonstrate that our proposed approach can effectively process the imbalanced data streams with concept drift.

## Experimental setup and parameters setting

To evaluate the effectiveness of the proposed method, we summarize and compare five methods of processing data stream samples. The specific parameters of the comparison algorithm are as follows: The number of nearest neighbors for REA is 10 and the post-balance ratio is 0.5. For EDAC, the error rate $x_i$ is the number of classification errors divided by the total number of samples. The threshold to determine sample quality $x_i$ is 0.1. We adopt UOB as the data preprocessing method of DWM and DWMIL. The ADAB and MWMOTEB are taking bagging as the basic framework, ADA, and MWMOTE are used as sampling methods separately. The number of partitions for each data chunk T of DWMIL, ADAB, MWMOTEB is 11. Update T value every ten data chunks for EDAC. The classification and regression tree (CART) as the learner of the base classifier for all methods, and for DWMIL and EDAC methods, the threshold for removing the dated classifier $\theta$ is set at 0.001. We detect concept drift instances by the CBCE algorithm. All experimentals are iterated 10 times to get the average result. All the other compared approaches have used the default parameter. We adopt a test-then-train strategy to evaluate the performance of the methods on each chunk, and use Recall and Area Under Curve (AUC) as the evaluation metrics to compare all methods. All these methods are implemented on an intel i5–3230m CPU, 4G RAM, and a single NVIDIA 610 GPU with MATLAB 2016a.

In the experiments, we select four synthetic data streams: SEA, Moving Gaussian, Hyper Plane, Checkerboard, and one real-world data stream: Electricity(All data streams can be obtained in the attachment). The percentage of the minority class samples in each chunk is 5% of the chunk size. It can be known that only 5% of the examples inside each data chunk belong to the minority class. SEA has three attributes randomized in [0, 10]. Where the first two attributes determine the labels of the data, the third attribute is irrelevantly treated as noise. We divided the SEA with 100000 instances into 100 data chunks. Moving Gaussian has two attributes and contains two classes of Normal distribution where the coordinates of the two classes are [5, 0] and [7, 0] gradually move to [-5,0] and [-3,0]. We divided the Moving Gaussian with 50,000 instances into 50 data chunks. Hyper Planes concept of data changes gradually $x_i$, where the attributed is 10, $x_i$ is used to decision hyperplane. And it was divided into 50 data chunks. Checkerboard is selected from the rotating chessboard, which has two attributes and 60000 instances and was divided into 200 data chunks. Electricity is the electricity price fluctuation *up/down* affected by demand and supply of the market from New South Wales, Australian. We divided the Electricity with 27,549 instances with seven attributes into 56 data chunks.

## Compared algorithms

We have compared the proposed algorithm with five state-of-the-art algorithms: REA, DWMIL, DWM, ADAB and MWMOTEB. We hereby give the general idea of each algorithm is as follows:

- REA: The REA algorithm selects the previous minority class samples that have some correlations with the current minority class samples to make the data chunks achieve balance. Training a classifier for each balanced data chunk, and combined together as an ensemble classifier in a dynamically weighted manner. Time Complexity: $O(n * m)$ for selecting relevant samples, where n is the number of previous minority samples and m is the number of current minority samples. Space Complexity: $O(n)$ for storing the historical minority samples. Ensemble Complexity: REA trains multiple classifiers on balanced chunks and combines them into an ensemble. Time Complexity: $O(c * (n + m))$ for training c classifiers on balanced chunks. Space Complexity: $O(c)$ for storing the ensemble.

- DWMIL: The DWMIL algorithm adopts UOB method to balance the data chunk, the initial weight is fixed to 1, and the weight of the classifier is adjusted dynamically according to the performance of the classifier. Finally, combined together as an ensemble classifier. Balancing Complexity: Uses the Unordered Oversampling Bagging (UOB) method to balance the data. UOB involves creating bags of samples from the original dataset, which is less complex than more sophisticated balancing techniques. Time Complexity: $O(b * (n + m))$, where b is the number of bags. Space Complexity: $O(b * (n + m))$. Ensemble Complexity: Adjusts the weights of the classifiers based on their performance, then combines them into an ensemble. Time Complexity: $O(c * (n + m))$ for training c classifiers. Space Complexity: $O(c)$.

- DWM: In DWM algorithm, the rule for classification on base classifier has been kept in a unchanged state. And the weight on base classifier is dynamically modified on each arrival data accordingly. Classification Complexity: Keeps the classification rule constant but dynamically modifies the weights of base classifiers. Time Complexity: $O(n + m)$ for updating weights and making predictions. Space Complexity: $O(c)$ for storing the classifiers and their weights.

- ADAB: The ADAB algorithm adopts ADASYN method to balance the data chunk. For each balanced data chunk, training a classifier and combined together as an ensemble classifier in a dynamically weighted manner. Balancing Complexity: Uses ADASYN (Adaptive Synthetic Sampling) to balance the data. ADASYN generates synthetic samples near the minority class samples, which can be computationally expensive.Time Complexity: $O(n^2)$ for generating synthetic samples, where n is the number of minority samples. Space Complexity: $O(n)$ for storing the synthetic samples. Ensemble Complexity: Trains classifiers on balanced data and combines them into an ensemble. Time Complexity: $O(c * (n + m))$. Space Complexity: $O(c)$.

- MWMOTEB: The MWMOTEB algorithm implements data chunk balance by using MWMOTE method. Then training a classifier for each data chunk. Finally, combined together as an ensemble classifier in a dynamically weighted manner. Balancing Complexity: Uses MWMOTE (Modified Wilson's Minority Oversampling Technique) to balance the data. MWMOTE is similar to ADASYN but may involve different steps for generating synthetic samples. Time Complexity: $O(n^2)$. Space Complexity: $O(n)$. Ensemble Complexity: Similar to ADAB. Time Complexity: $O(c *(n + m))$. Space Complexity: $O(c)$.

## Comparison and analysis

The AUC, G-mean, and Recall are used as evaluation indexes, and the strategy of testing before training is used to evaluate various algorithms on each data chunk. The average overall prediction accuracy of the comparative algorithms is shown in Tables 1–3. Under the metric of AUC by comparing with other algorithms, the EDAC algorithm has the better performance in Moving Gaussian, Hyperplane, Checkerboard, and Electricity. The best performance is in the

**Table 1. The average overall prediction accuracy of different algorithms in AUC measurement.**

| Data | REA | DWMIL | DWM | ADAB | MWMOTEB | EDAC |
|---|---|---|---|---|---|---|
| Moving Gaussian | 0.774 | 0.855 | 0.801 | 0.840 | 0.781 | 0.867 |
| SEA | 0.978 | 0.976 | 0.923 | 0.989 | 0.983 | 0.981 |
| Hyper plane | 0.638 | 0.680 | 0.565 | 0.607 | 0.549 | 0.698 |
| Checkerboard | 0.839 | 0.886 | 0.609 | 0.882 | 0.873 | 0.893 |
| Electricity | 0.773 | 0.830 | 0.619 | 0.813 | 0.785 | 0.830 |

**Table 2. The average overall prediction accuracy of different algorithms in gm measurement.**

| Data | REA | DWMIL | DWM | ADAB | MWMOTEB | EDAC |
|---|---|---|---|---|---|---|
| *Moving Gaussian* | 0.219 | 0.760 | 0.783 | 0.812 | 0.543 | 0.800 |
| *SEA* | 0.893 | 0.920 | 0.905 | 0.935 | 0.922 | 0.921 |
| *Hyper plane* | 0.336 | 0.589 | 0.550 | 0.404 | 0.394 | 0.616 |
| *Checkerboard* | 0.757 | 0.811 | 0.596 | 0.815 | 0.789 | 0.823 |
| *Electricity* | 0.621 | 0.707 | 0.587 | 0.622 | 0.524 | 0.718 |

**Table 3. The average overall prediction accuracy of different algorithms in recall measurement.**

| Data | REA | DWMIL | DWM | ADAB | MWMOTEB | EDAC |
|---|---|---|---|---|---|---|
| *Moving Gaussian* | 0.111 | 0.650 | 0.637 | 0.515 | 0.333 | 0.661 |
| *SEA* | 0.868 | 0.936 | 0.896 | 0.934 | 0.908 | 0.956 |
| *Hyper plane* | 0.148 | 0.478 | 0.561 | 0.207 | 0.212 | 0.628 |
| *Checkerboard* | 0.709 | 0.771 | 0.674 | 0.726 | 0.676 | 0.816 |
| *Electricity* | 0.557 | 0.679 | 0.650 | 0.462 | 0.358 | 0.729 |

Checkerboard data stream, which increases 28% compared with the DWM algorithm. On the SEA data stream, the values of the EDAC algorithm are basically the same as the other five algorithms. For the G-mean evaluation criteria, the best performance is on the Hyper Plane data stream, 28% higher than that of the REA algorithm. Because we increase the recognition accuracy of samples by introducing effective information content. The Recall value of the EDAC algorithm is obviously superior to other algorithms. In which, the best performance is in Moving Gaussian data stream, which increases 55% compared with REA algorithm.

EDAC algorithm dynamically updates the weight of the classifier, through the evaluation index value, and constantly eliminates the classifier with a lower weight. Compared with the DWM algorithm and DWMIL algorithm that only uses a single evaluation index to update the classifier by Single method that reduces the weight of the classifier through the increase of time. The number of negative samples identified in all negative samples is larger than other algorithms, and because the recall value is better than other algorithms, the g-mean value is better than other algorithms; AUC value is the area enclosed by the ROC curve and coordinate axis. EDAC improves the recognition accuracy of positive samples and negative samples through various mechanisms, making the model more robust, and the area under ROC offline is larger than other algorithms, finally making AUC value better than other algorithms. In general, the EDAC algorithm has better performance than other algorithms in dealing with imbalanced data streams with conceptual drift.

The AUC and Recall performance on each chunk are show in Figs 3–12. Since the REA, ADAB, MWMOTEB, DWMIL, EDAC use the same base classifier, they all have roughly the same initial value. For the metric of AUC, the EDAC algorithm shows excellent performance in various data streams. ADAB is almost as good as MWMOTEB, since they adopt a similar sampling mechanism and ensemble algorithm. On average, the DWM algorithm is the worst of all algorithms, DWMIL algorithm is slightly better than the REA algorithm. EDAC and DWMIL both have a good ability to resist concept drift. But the performance of EDAC is higher than that of DWMIL. They adopt a similar classifier weight update strategy, but EDAC increases the recognition accuracy of minority instances in the sampling process.

For Recall, the CBCE algorithm keeps updating itself to adapt to the continuous concept drift of Moving Gaussian data stream, so the identification of minority instances is low. The

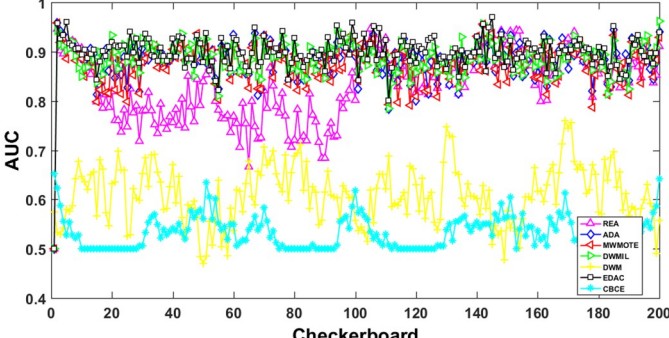

**Fig 3. Illustration for AUC performance of each chunk on data streams Checkerboard.**

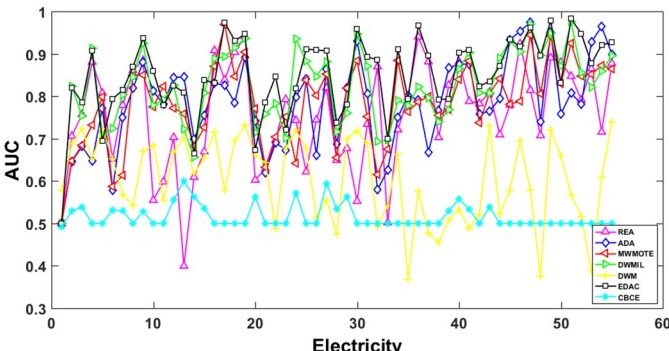

**Fig 4. Illustration for AUC performance of each chunk on data streams Electricity.**

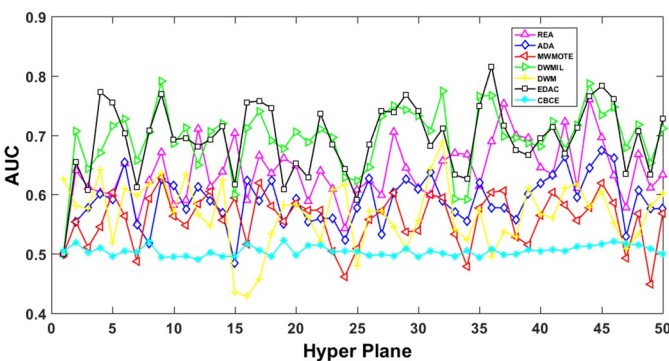

**Fig 5. Illustration for AUC performance of each chunk on data streams Hyper Plane.**

results about Moving Gaussian show that CBCE predicts all instances as negative and therefore receives 0 in Recall. REA algorithm stores the minority instances in the previous data chunk. But due to concept drift, majority instances would be considered as minority instances in some data chunks. It could lead to the performance of algorithm REA which is possible to introduce noise instances into classifier training getting worse and worse. Therefore, it

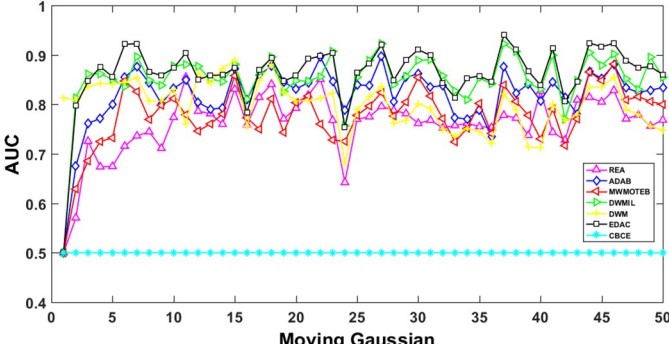

**Fig 6. Illustration for AUC performance of each chunk on data streams Moving Gaussian.**

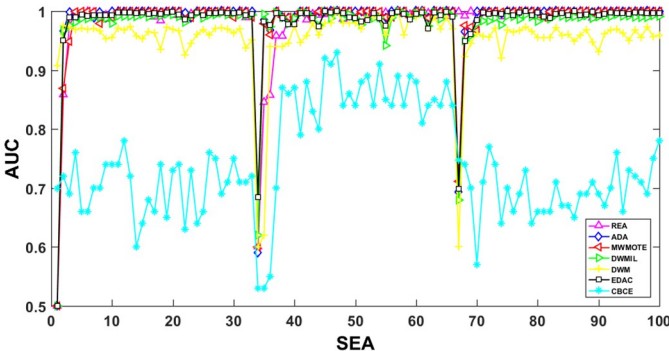

**Fig 7. Illustration for AUC performance of each chunk on data streams SEA.**

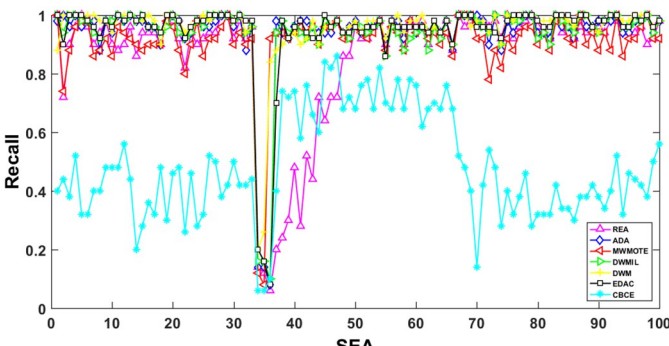

**Fig 8. Illustration for Recall performance of each chunk on data SEA.**

gradually drops to 0. It is obviously shown in the SEA data stream that the recovery of algorithm REA is the slowest when the concept changes greatly.

Generally speaking, EDAC recovers its better performance quickly compared with other methods. DWMIL also responds quickly, but its recognition accuracy of minority samples is far less than that of the EDAC algorithm. DWM needs to spend about 10 chunks to recover

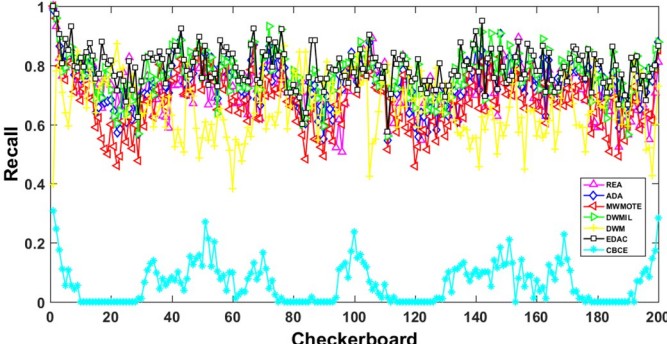

**Fig 9. Illustration for Recall performance of each chunk on data streams Checkerboard.**

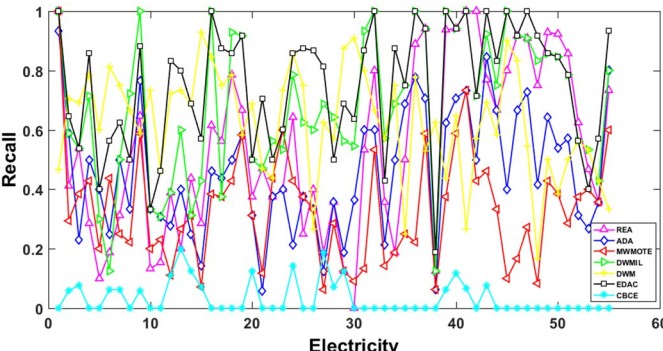

**Fig 10. Illustration for Recall performance of each chunk on data streams Electricity.**

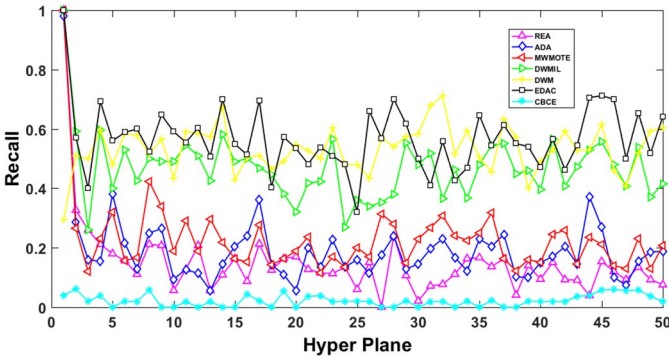

**Fig 11. Illustration for Recall performance of each chunk on data streams Hyper Plane.**

the performance. ADAB and MWMOTE use an oversampling algorithm to balance data chunk, but the classifier they trained in the current data chunk was used to evaluate the data chunk arrived later. In this case, the performance of the sampling method is poor, even worse than the DWM algorithm which does not process anything. These three indicators evaluate the performance of classification models from different perspectives. AUC provides

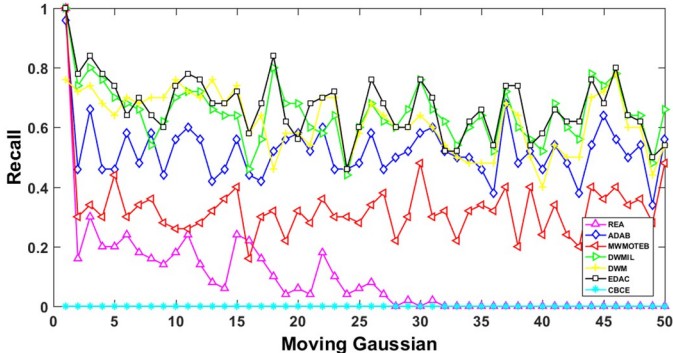

**Fig 12. Illustration for Recall performance of each chunk on data streams Moving Gaussian.**

information about the overall discriminative ability of the classifier, G-Means helps balance accuracy and recall, especially in cases of imbalanced categories, while Recall focuses particularly on the completeness of positive example recognition. Our algorithm EDAC may have lower GM values than other algorithms in certain specific situations. Overall, our algorithm EDAC has achieved excellent performance in all three indicators. In the later stage, we will increase the number of sample points and sacrifice a small amount of time to increase the accuracy of the algorithm.

## Results of time complexity

Table 4 shows the running time of the six algorithms in the five data streams. It can be known that the cost of EDAC algorithm is less than the other methods on four data streams. The reason is that EDAC adopts dynamically bagging with twice lower than fixed T decision trees Compared with DWMIL. And in the process of sampling, we fixed selection a part of high quality minority class samples, which saves time in each basic classifier training compared with the UOB algorithm. REA maintains a single decision tree for each chunk, and keep all the decision tree trained in each data chunk. This method is suitable for small-scale samples. Thus, REA is faster for the Electricity data stream which has a smaller number of instances and chunks. However, it uses the KNN algorithm to select similar samples from all the previous minority class samples to the current class. As the number of chunks increases, the number of calculations increases. On the Checkerboard which contains 200 chunks, the cost of EDAC is obviously lower than that of REA. The running cost of algorithm DWM is the largest for all compared algorithm, which updates the ensemble classifiers on every incoming instance where other algorithms are based on data chunks, and the processing time will be faster than this one-to-one learning method. ADAB, MWMOTEB and DWMIL use the same data chunk processing method, but the sampling method is different, where ADAB adopts ADASYN

**Table 4. List of the time used by each algorithm.**

| Data | REA | DWMIL | DWM | ADAB | MWMOTEB | EDAC |
|---|---|---|---|---|---|---|
| Moving Gaussian | 11.9 | 10.6 | 133.6 | 15.9 | 25.0 | 8.0 |
| SEA | 45.6 | 32.8 | 224.2 | 36.6 | 59.5 | 24.3 |
| Hyper plane | 14.5 | 10.9 | 383.3 | 19.2 | 43.7 | 10.4 |
| Checkerboard | 282.7 | 54.6 | 272.3 | 78.7 | 176.9 | 50.8 |
| Electricity | 4.8 | 12.8 | 95.7 | 16.3 | 17.2 | 7.9 |

sampling method, MWMOTEB applies MWMOTE sampling method, DWMIL exploits UOB sampling method. Therefore, the time spent by ADAB and MWMOTEB is not much different, while the time spent by DWMIL is relatively less.

## Conclusions

In this paper, we investigate the problem of imbalanced data stream with concept drift. The EDAC algorithm has been proposed to tackle such issue. The main idea of this EDAC is to leverage the entropy value of data samples to calculate the effective information content of samples. Based on the gained effective information above, EDAC divides the imbalanced data chunk into multiple balanced sample pairs. Next, with aim of developing the capability of decision boundary, the base classifier is built upon balanced sample pairs by introducing the effective information content of samples into classifier creation. Also, the weight on each based classifier is calculated and updated based on forthcoming data chunks. Last, we have conducted the simulation via synthetic data streams but also in real-world data streams. Compared to the previous work, the results have shown that the proposed algorithm has the best performance for improvement of the classification accuracy towards the imbalanced data stream with concept drift.

## Supporting information

**S1 Dataset.**
(XLSX)

**S2 Dataset.**
(XLSX)

**S3 Dataset.**
(XLSX)

**S4 Dataset.**
(XLSX)

**S5 Dataset.**
(XLSX)

**S1 File.**
(DOCX)

**S1 Data.**
(ZIP)

## Author Contributions

**Conceptualization:** JiaMing Gong.

**Data curation:** JiaMing Gong.

**Formal analysis:** JiaMing Gong.

**Funding acquisition:** JiaMing Gong.

**Investigation:** JiaMing Gong.

**Methodology:** JiaMing Gong.

**Project administration:** JiaMing Gong.

**Resources:** JiaMing Gong.

**Software:** JiaMing Gong.

**Supervision:** JiaMing Gong.

**Validation:** JiaMing Gong.

**Visualization:** JiaMing Gong.

**Writing – original draft:** JiaMing Gong.

**Writing – review & editing:** MingGang Dong.

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
