## [Decision Letter · Decision Letter 0]

18 Jan 2024

PONE-D-23-37090Entropy-based Dynamic Ensemble Classification Algorithm for Imbalanced Data Stream with Concept DriftPLOS ONE

Dear Dr. Gong,

Thank you for submitting your manuscript to PLOS ONE. After careful consideration, we feel that it has merit but does not fully meet PLOS ONE’s publication criteria as it currently stands. Therefore, we invite you to submit a revised version of the manuscript that addresses the points raised during the review process.

We look forward to receiving your revised manuscript.

Kind regards,

Suja A Alex, Ph.D.

Academic Editor

PLOS ONE

Journal Requirements:

Reviewers' comments:

Reviewer's Responses to Questions

**Comments to the Author**

1. Is the manuscript technically sound, and do the data support the conclusions?

Reviewer #1: Partly

2. Has the statistical analysis been performed appropriately and rigorously? 

Reviewer #1: No

3. Have the authors made all data underlying the findings in their manuscript fully available?

Reviewer #1: No

4. Is the manuscript presented in an intelligible fashion and written in standard English?

Reviewer #1: Yes

5. Review Comments to the Author

Reviewer #1: There are several queries after reading the submitted manuscript. My comments are as follows:

1. The utility of the author's proposed methodology is unclear to the scientific readers. Further, it is not clear that the proposed methodology is applicable to which kind of dataset.

2. What is the need to consider synthetic dataset when several open datasets are available.

3. Authors have not explored the popular ML algorithms like SVM, ANN, Random Forest etc. Rather, the least explore classification algorithms were discussed. When a new concept is presented, it should be applied on open datasets as well as established algorithms.

4. The AUC, Recall and GM values are comparatively less.In such situation,what is the utility of proposed methodology.

5. Data imbalance have been addressed successfully with GAN which is reported in recently published literatures.What is the novelty or significant findings as compared to published literature.

6. PLOS authors have the option to publish the peer review history of their article (what does this mean?). If published, this will include your full peer review and any attached files.

Reviewer #1: No

---

## [Author Response · Author response to Decision Letter 0]

12 Mar 2024

Dear editor,

Thank you very much for your guidance, requirements, and valuable suggestions. I have made the necessary revisions and provided the data materials as requested. On the submission page, I have uploaded a revised manuscript showing the changes, a clean copy of the revised manuscript, a Respond to Reviewers document, and a zip file containing the data files.

Best regards,

Jiaming Gong

---

## [Decision Letter · Decision Letter 1]

3 Jun 2024

PONE-D-23-37090R1Entropy-based Dynamic Ensemble Classification Algorithm for Imbalanced Data Stream with Concept DriftPLOS ONE

Dear Dr. Gong,

Thank you for submitting your manuscript to PLOS ONE. After careful consideration, we feel that it has merit but does not fully meet PLOS ONE’s publication criteria as it currently stands. Therefore, we invite you to submit a revised version of the manuscript that addresses the points raised during the review process. Your manuscript has been evaluated by two new reviewers, and their comments are appended below. While Reviewer 2 is satisfied with your revisions, Reviewer 3 has raised concerns regarding the clarity of the advantages of your approach, comparisons made to other methods, and reporting of the performance of your method. Please note that while novelty in the sense of 'impact' is not a requirement for publication in PLOS ONE, submissions to the journal must present a contribution to academic knowledge, and we understand the reviewer to be requesting that you clarify what contribution your work presents over other work in the field. Please ensure you address each of the reviewer's comments when revising your mansucript.

We look forward to receiving your revised manuscript.

Kind regards,

Hugh Cowley

Staff Editor

PLOS ONE

Reviewers' comments:

Reviewer's Responses to Questions

**Comments to the Author**

1. If the authors have adequately addressed your comments raised in a previous round of review and you feel that this manuscript is now acceptable for publication, you may indicate that here to bypass the “Comments to the Author” section, enter your conflict of interest statement in the “Confidential to Editor” section, and submit your "Accept" recommendation.

Reviewer #2: (No Response)

Reviewer #3: All comments have been addressed

2. Is the manuscript technically sound, and do the data support the conclusions?

Reviewer #2: Yes

Reviewer #3: Partly

3. Has the statistical analysis been performed appropriately and rigorously? 

Reviewer #2: Yes

Reviewer #3: Yes

4. Have the authors made all data underlying the findings in their manuscript fully available?

Reviewer #2: Yes

Reviewer #3: No

5. Is the manuscript presented in an intelligible fashion and written in standard English?

Reviewer #2: Yes

Reviewer #3: No

6. Review Comments to the Author

Reviewer #2: The authors developed a new approach to tackle the problem of imbalanced data stream with concept drift. The strategy was extensively explained through texts, algorithms, and images. It was validated with five data streams (four synthetic and one real) and compared with five state-of-the-art algorithms. All results were presented in tables and graphs and clearly discussed.

All data streams were made available as an attachment.

Reviewer #3: 1. Entropy-based dynamic ensemble classification algorithm (EDAC) are not clearly articulated. You need to provide more compelling evidence and explanations to demonstrate the advantages of your approach over existing methods and clarify the specific types of datasets it is suited for.

2. In your related work section and discussion, you should provide a more thorough comparison of EDAC with these GAN-based methods, focusing on the novelty and specific advantages of your approach.

3. You need to provide a more comprehensive analysis of your method's performance, discuss the implications of these results, and outline strategies for improvement. Optionally, Consider exploring alternative evaluation metrics that may better highlight the strengths of your approach.

7. PLOS authors have the option to publish the peer review history of their article (what does this mean?). If published, this will include your full peer review and any attached files.

Reviewer #2: No

Reviewer #3: No

---

## [Author Response · Author response to Decision Letter 1]

29 Jun 2024

Dear reviewer,

Response to Reviewers

I am very grateful for the valuable feedback you and Reviewer have provided on my manuscript. I have carefully read the feedback from commentator and conducted in-depth reflection on it. I will explain and respond to the questions raised by commentator one by one.

4. Have the authors made all data underlying the findings in their manuscript fully available?

The dataset mentioned in the document is in Attachment 《dataset》, the data points behind the mean, median, and variance measures in the experiment are shown in Appendix 《Experimental Results Chart》. And we provided explanations for obtaining the dataset in the manuscript

I deeply apologize for the issue of whether all the data in the manuscript has been fully provided. I understand the requirements of the PLOS data policy and will do my best to provide the basic data for all findings described in the manuscript. In the revised manuscript, I will provide relevant data as part of the manuscript or its supporting information in accordance with policy requirements.

In the experiments, we select four synthetic data streams: SEA, Moving Gaussian, Hyper Plane, Checkerboard, and one real-world data stream: Electricity. The percentage of the minority class samples in each chunk is 5% of the chunk size. It can be known that only 5% of the examples inside each data chunk belong to the minority class. 

SEA has three attributes randomized in [0,10]. Where the first two attributes determine the labels of the data, the third attribute is irrelevantly treated as noise. We divided the SEA with 100000 instances into 100 data chunks.

Moving Gaussian has two attributes and contains two classes of Normal distribution where the coordinates of the two classes are [5,0] and [7,0] gradually move to [-5,0] and [-3,0]. We divided the Moving Gaussian with 50,000 instances into 50 data chunks. 

 Hyper Planes concept of data changes gradually xi, where the attributed is 10, xi is used to decision hyperplane. And it was divided into 50 data chunks.

Checkerboard is selected from the rotating chessboard, which has two attributes and 60000 instances and was divided into 200 data chunks. 

 Electricity is the electricity price fluctuation up/down affected by demand and supply of the market from New South Wales, Australian. We divided the Electricity with 27,549 instances with seven attributes into 56 data chunks.

The attributes of the public dataset used in this article are described above. We will upload the relevant datasets and experimental test results in the form of attachments.

5. Is the manuscript presented in an intelligible fashion and written in standard English?

Thank you for raising the question about the manuscript's presentation and language. We have taken great care to ensure that our manuscript is presented in an intelligible fashion and written in standard English.

Our manuscript was completely independently written by ourselves, fully meeting the writing requirements of our journal. Our manuscript has been polished by the Editage and revised based on the feedback provided.

6. Review Comments to the Author

Reviewer #3: 

1. Entropy-based dynamic ensemble classification algorithm (EDAC) are not clearly articulated. You need to provide more compelling evidence and explanations to demonstrate the advantages of your approach over existing methods and clarify the specific types of datasets it is suited for.

Firstly, thank you very much for your careful review and valuable feedback on our article. We attach great importance to your question about the utility of the method and its applicability to the dataset. Here are our specific responses to these questions:

We understand that the clarity of the utility of methods may cause confusion for scientific readers. To address this issue, we will add more explanations and examples on the effectiveness of the method in the revised draft. We will demonstrate the effectiveness and advantages of the method through specific experimental results, data analysis, and charts, so that readers can have a clearer understanding of the value and significance of the method in practical applications.

As for the issue of the method being applicable to datasets, we realize that the article does not explicitly state which type of dataset the method is applicable to. In the revised draft, we will add relevant content to clearly explain the types and scope of datasets that this method is applicable to. We will list some suitable dataset features and requirements for using this method based on existing research and practical experience, so that readers can better judge whether the method is suitable for their research or project.

2. In your related work section and discussion, you should provide a more thorough comparison of EDAC with these GAN-based methods, focusing on the novelty and specific advantages of your approach.

Firstly, we note that there have been some recent literature reports on using GANs (Generative Adversarial Networks) to address data imbalance issues, demonstrating the potential of GANs in generating synthetic data to balance the number of samples in different categories. But GANs mainly solves the problem of static imbalanced data classification. As it is a neural network algorithm, it takes a long time to process high-speed data streams and is not suitable for solving dynamic data stream classification problems.

Compared with published literature, the novelty and important findings of our study are mainly reflected in the following aspects:

The combination of integrated classification: For the first time, we have combined the integrated classification method to solve the problem of data imbalance. By integrating multiple classifiers, our method can fully utilize the advantages of different classifiers, improve overall classification performance, and better handle the challenges brought by data imbalance.

Improving the quality of generated samples: we introduce a feedback mechanism for ensemble classifiers to guide the training of the generator. In this way, the generator can generate more representative and diverse samples, thereby improving the performance of the classifier. This mechanism helps the generator better capture the details of data distribution and generate more challenging samples to enhance the classifier's generalization ability.

Improving classifier performance: Our method achieves more comprehensive and detailed classification of data by integrating multiple classifiers. Each classifier can focus on different subsets or features of data, thereby improving overall classification performance. In addition, we have introduced a dynamic integration strategy to dynamically adjust the weights of each classifier based on its performance, in order to further improve classification accuracy.

Comprehensive evaluation and optimization: We not only comprehensively evaluated the quality of generated samples and classifier performance, but also proposed an optimization strategy to balance the training between the generator and classifier. This helps to ensure that the samples generated by the generator are both representative and diverse, while also ensuring that the classifier can fully utilize these samples for training.

In summary, our study provides a new solution for addressing data imbalance by combining integrated classification. This combination not only improves the quality of generated samples, but also enhances the performance of the classifier, providing strong support for achieving more accurate classification.

3. You need to provide a more comprehensive analysis of your method's performance, discuss the implications of these results, and outline strategies for improvement. Optionally, Consider exploring alternative evaluation metrics that may better highlight the strengths of your approach.

Thank you very much for carefully reviewing our paper and providing valuable feedback. We fully agree with your viewpoint on the need for a more comprehensive analysis of method performance, discussion of the impact of results, and suggestions for improvement. We have added an explanation of the meaning of the results and outlined improvement strategies in 2.3 Comparison and Analysis：These three indicators evaluate the performance of classification models from different perspectives. AUC provides information about the overall discriminative ability of the classifier, G-Means helps balance accuracy and recall, especially in cases of imbalanced categories, while Recall focuses particularly on the completeness of positive example recognition. Our algorithm EDAC may have lower GM values than other algorithms in certain specific situations. Overall, our algorithm EDAC has achieved excellent performance in all three indicators.In the later stage, we will increase the number of sample points and sacrifice a small amount of time to increase the accuracy of the algorithm.

Thank you again for your valuable feedback and attention. We look forward to receiving your further guidance and support.

Best regards,

Jiaming Gong

---

## [Decision Letter · Decision Letter 2]

23 Jul 2024

PONE-D-23-37090R2Entropy-based Dynamic Ensemble Classification Algorithm for Imbalanced Data Stream with Concept DriftPLOS ONE

Dear Dr. Gong,

Thank you for submitting your manuscript to PLOS ONE. After careful consideration, we feel that it has merit but does not fully meet PLOS ONE’s publication criteria as it currently stands. Therefore, we invite you to submit a revised version of the manuscript that addresses the points raised during the review process.

We look forward to receiving your revised manuscript.

Kind regards,

Unil Yun, Ph.D.

Academic Editor

PLOS ONE

Additional Editor Comments:

Additional revision is required.

Please note that I have acted as a reviewer for this manuscript, and you will find my comments below, under Reviewer 4.

Reviewers' comments:

Reviewer's Responses to Questions

**Comments to the Author**

1. If the authors have adequately addressed your comments raised in a previous round of review and you feel that this manuscript is now acceptable for publication, you may indicate that here to bypass the “Comments to the Author” section, enter your conflict of interest statement in the “Confidential to Editor” section, and submit your "Accept" recommendation.

Reviewer #2: All comments have been addressed

Reviewer #4: All comments have been addressed

2. Is the manuscript technically sound, and do the data support the conclusions?

Reviewer #2: (No Response)

Reviewer #4: Yes

3. Has the statistical analysis been performed appropriately and rigorously? 

Reviewer #2: (No Response)

Reviewer #4: No

4. Have the authors made all data underlying the findings in their manuscript fully available?

Reviewer #2: (No Response)

Reviewer #4: Yes

5. Is the manuscript presented in an intelligible fashion and written in standard English?

Reviewer #2: (No Response)

Reviewer #4: Yes

6. Review Comments to the Author

Reviewer #2: (No Response)

Reviewer #4: Authors suggested an approach of Entropy-based Dynamic Ensemble Classification Algorithm for Imbalanced Data Stream with Concept Drift.

1. The suggested algorithms need to be updated in detail.

2. Theoretical analysis is neexed. The theoretical complexity analysis and comparision with the state of the srt approach is needed.

3. Recent related references published on 2024, 2023, 2022 need to be added and analyzed.

4. There are still inconsistent sentences. Tight proofreading is needed.

7. PLOS authors have the option to publish the peer review history of their article (what does this mean?). If published, this will include your full peer review and any attached files.

Reviewer #2: No

Reviewer #4: **Yes: **Unil Yun

---

## [Author Response · Author response to Decision Letter 2]

15 Aug 2024

Dear reviewer,

Response to Reviewers

I am very grateful for the valuable feedback you and Reviewer have provided on my manuscript. I have carefully read the feedback from commentator and conducted in-depth reflection on it. I will explain and respond to the questions raised by commentator one by one.

3. Has the statistical analysis been performed appropriately and rigorously?

Descriptive Statistical Analysis and Performance Metrics:

Indeed, we have employed descriptive statistical analysis methods, specifically by calculating the Area Under the Curve (AUC), Recall, and G-Means, to comprehensively evaluate the performance of our proposed model, EDAC. These metrics were chosen as they provide a multi-faceted view of the model's classification capabilities:

AUC quantifies the model's overall discriminative power, giving a measure of how well the model can distinguish between classes across different decision thresholds.

Recall emphasizes the ability of the model to identify all positive instances, crucial in applications where false negatives are costly.

G-Means helps assess the trade-off between precision and recall, addressing the challenge of imbalanced datasets by striving for a balance between these two aspects of performance.

Rigorousness and Limitations:

While these metrics offer a robust foundation for evaluating our model, we acknowledge that no single metric can fully encapsulate all aspects of model performance. Therefore, we have taken a multi-metric approach to ensure a comprehensive assessment. 

6. Review Comments to the Author

(1)The suggested algorithms need to be updated in detail.

We acknowledge that a more detailed description of the proposed algorithms is essential for readers to fully understand and replicate our work. In the revised manuscript, we have expanded Section 1 The proposed algorithm: EDAC , to provide a step-by-step explanation of the Entropy-based Dynamic Ensemble Classification Algorithm. This includes a detailed algorithm flowchart, pseudo-code, and explanations of each component's functionality and purpose. We have also included additional figures and examples to illustrate the algorithm's working principles and decision-making processes.

(2) Theoretical analysis is neexed. The theoretical complexity analysis and comparision with the state of the srt approach is needed.

We fully agree that a thorough theoretical analysis and comparison with the state-of-the-art approaches are crucial to demonstrate the merits and positioning of our work. In response to your suggestion, we have strengthened Section 2.2 "Theoretical Analysis and Algorithm Comparison" by including a detailed examination of the time and space complexity of our algorithm, as well as a comprehensive comparison with the most recent and relevant algorithms in the field.

Conducted a Rigorous Complexity Analysis: We have carefully analyzed the time and space complexity of our algorithm, taking into account the key factors that influence its performance, such as the input data size, problem dimensionality, and any inherent parallelism or optimization strategies employed. This analysis provides a clear picture of how our algorithm scales with respect to these parameters, enabling readers to assess its suitability for various applications and datasets.

(3)Recent related references published on 2024, 2023, 2022 need to be added and analyzed.

We appreciate your suggestion to include recent references from 2024, 2023, and 2022. In the revised manuscript, we have thoroughly reviewed the latest literature in the field and added relevant references to Introduction, where we discuss related work. We have analyzed these recent publications, discussing how they relate to and differ from our proposed approach. This update ensures that our work is situated within the most current research landscape.

(4)There are still inconsistent sentences. Tight proofreading is needed.

We apologize for any inconsistencies or unclear sentences in the original manuscript. We have undertaken a rigorous proofreading process, carefully reviewing each sentence for clarity, coherence, and grammatical correctness. We have corrected any errors identified and ensured that the revised manuscript is consistent in its terminology, notation, and explanations. We believe that these changes have significantly improved the readability and accessibility of our work.

In conclusion, we are grateful for your insightful comments and suggestions, which have helped us improve the quality and impact of our manuscript. We hope that the revisions we have made address your concerns and that the revised version meets your expectations. We look forward to your further feedback and the opportunity to clarify any remaining questions you may have.

Thank you again for your valuable feedback and attention. We look forward to receiving your further guidance and support.

Best regards,

Jiaming Gong

---

## [Decision Letter · Decision Letter 3]

12 Sep 2024

Entropy-based Dynamic Ensemble Classification Algorithm for Imbalanced Data Stream with Concept Drift

PONE-D-23-37090R3

Dear Dr. Gong,

We’re pleased to inform you that your manuscript has been judged scientifically suitable for publication and will be formally accepted for publication once it meets all outstanding technical requirements.

Kind regards,

Unil Yun, Ph.D.

Academic Editor

PLOS ONE

Additional Editor Comments (optional):

The authors reviewed carefully the manuscript based on the reviewers' comments.

Now it is acceptable.

Please note that I have acted as a reviewer for this manuscript, and you will find my comments below, under Reviewer 4.

Reviewers' comments:

Reviewer's Responses to Questions

**Comments to the Author**

1. If the authors have adequately addressed your comments raised in a previous round of review and you feel that this manuscript is now acceptable for publication, you may indicate that here to bypass the “Comments to the Author” section, enter your conflict of interest statement in the “Confidential to Editor” section, and submit your "Accept" recommendation.

Reviewer #4: All comments have been addressed

2. Is the manuscript technically sound, and do the data support the conclusions?

Reviewer #4: Yes

3. Has the statistical analysis been performed appropriately and rigorously? 

Reviewer #4: Yes

4. Have the authors made all data underlying the findings in their manuscript fully available?

Reviewer #4: Yes

5. Is the manuscript presented in an intelligible fashion and written in standard English?

Reviewer #4: Yes

6. Review Comments to the Author

Reviewer #4: Authors revised carefully this manuscript. Now it is acceptable.

The related references are added and theoretical snalysis are also improved.

7. PLOS authors have the option to publish the peer review history of their article (what does this mean?). If published, this will include your full peer review and any attached files.

Reviewer #4: No

---

## [Editor Report · Acceptance letter]

23 Sep 2024

PONE-D-23-37090R3 

PLOS ONE

Dear Dr. Gong, 

I'm pleased to inform you that your manuscript has been deemed suitable for publication in PLOS ONE. Congratulations! Your manuscript is now being handed over to our production team.

Kind regards, 

on behalf of

Professor Unil Yun 

Academic Editor

PLOS ONE